# IMU Motion Capture Method with Adaptive Tremor Attenuation in Teleoperation Robot System

**DOI:** 10.3390/s22093353

**Published:** 2022-04-27

**Authors:** Huijin Zhu, Xiaoling Li, Long Wang, Zhangyi Chen, Yueyang Shi, Shuai Zheng, Min Li

**Affiliations:** 1School of Mechanical Engineering, Xi’an Jiaotong University, Xi’an 710000, China; zhhjjym@stu.xjtu.edu.cn (H.Z.); wangl521@stu.xjtu.edu.cn (L.W.); 3120101240@stu.xjtu.edu.cn (Z.C.); victories@stu.xjtu.edu.cn (Y.S.); min.li@mail.xjtu.edu.cn (M.L.); 2School of Software Engineering, Xi’an Jiaotong University, Xi’an 710000, China; shuaizheng@xjtu.edu.cn

**Keywords:** IMU, regression model, physiological tremor, EKF, sEMG signal, teleoperation system

## Abstract

Teleoperation robot systems can help humans perform tasks in unstructured environments. However, non-intuitive control interfaces using only a keyboard or joystick and physiological tremor reduce the performance of teleoperation. This paper presents an intuitive control interface based on the wearable device gForcePro+ armband. Two gForcePro+ armbands are worn at the centroid of the upper arm and forearm, respectively. Firstly, the kinematics model of the human arm is established, and the inertial measurement units (IMUs) are used to capture the position and orientation information of the end of the arm. Then, a regression model of angular transformation is developed for the phenomenon that the rotation axis of the torsion joint is not perfectly aligned with the limb segment during motion, which can be applied to different individuals. Finally, to attenuate the physiological tremor, a variable gain extended Kalman filter (EKF) fusing sEMG signals is developed. The described control interface shows good attitude estimation accuracy compared to the VICON optical capture system, with an average angular RMSE of 4.837° ± 1.433°. The performance of the described filtering method is tested using the xMate3 Pro robot, and the results show it can improve the tracking performance of the robot and reduce the tremor.

## 1. Introduction

Robotic manipulators can use teleoperation to interact with a wide array of objects and scenarios in unstructured environments [1]. Through teleoperation, human cognitive abilities can be used to assist robots in handling decisions that are particularly difficult for autonomy [2]. Robots have a wide range of applications and are now entering the daily life of human beings; it is becoming more likely that novices will work with robots rather than robotic experts [3,4]. In robot teleoperation, a major area of research focuses on providing a control interface that is intuitive and easily preserves situation awareness [5]. Hence, teleoperation can be accessed by novices with less training time. A teleoperation control interface can be visited with hand motions as opposed to joysticks and point-and-click interfaces, as they are natural movements to the teleoperator [6].

The optical tracking system includes multiple cameras and depth cameras. Multi-camera motion capture systems track active or passive markers attached to anatomical areas of the body to measure joint motion. Multi-camera is considered the measurement standard, providing very accurate motion estimates, and the results of other motion capture systems are often compared and validated against the multi-camera motion capture results [7]. The depth camera can detect the depth of field distance in the shooting space. Microsoft Kinect and Leap Motion controller are two different depth motion capture systems used for upper limb motion estimation [8]. Both systems use the human skeleton to track 3D motion without the use of any markers. When using a depth camera system, it is possible for the user to be occluded by other people or objects in the system’s field of view, and interactions that rely on human skeletal feature detection can be affected [9]. During the occlusion process, the system is unable to recognize the human joint position. Compared to multiple cameras, depth camera systems are less expensive and easier to move [10]. In general, optical tracking systems are not wearable and are not suitable for unstructured or outdoor scenarios.

sEMG signals reflect the level of muscle activation and can thereby be used to predict or identify human movement. Due to the nonlinear characteristics of the sEMG signal, the prediction of a single joint angle or position is easier than a multi joint angle or position, and better performance can be obtained [11]. Tang et al. used a back propagation neural network (BPN) for motion estimation of the elbow joint based on sEMG signals with a root mean square error (RMSE) of 10.93 [12]. However, it is difficult to capture multiple degrees of freedom of motion simultaneously. Furthermore, sEMG signals can be used to determine body stiffness and muscle activity to enhance teleoperation performance [13,14].

In contrast to the above methods, inertial measurement units (IMUs) have been widely used for motion tracking because they offer the advantages of being of low cost and small size and can be easily integrated with wearable devices for accurate, non-invasive and portable motion tracking. IMU provides information captured by multi-axis accelerometers, multi-axis gyroscopes and multi-axis magnetometers to determine the position and direction of human joints, with good accuracy for human motion estimation [15,16]. Caputo et al. developed a modular inertial motion capture system to estimate the orientation and posture angle trends of each body segment [17]. The great interest of researchers in the inertial measurement unit is mainly due to the fact that, compared to optical motion tracking systems, IMUs are not affected by occlusion and illumination conditions and allow for tracking of the entire body in an unstructured dynamic environment [18].

The IMU-based motion capture system can obtain the position and orientation of the arm, but since wearable devices with built-in IMUs are worn in the middle of the arm in most cases, there is a misalignment between the axis of rotation of the distorted joint and the limb segment during the rotational motion of the arm [19]. In addition, during teleoperation, there is always dynamic uncertainty in the robot due to sensor noise and physiological tremor of the arm; its tracking error increases with time and the robot jitters, which affects the reliability, control accuracy and stability of teleoperation and therefore requires a continuous filtering process [20,21].

Physiological tremor is a nonlinear stochastic process with amplitudes ranging from 50 to 100 µm and frequencies ranging from 6 to 15 Hz in each principal axis, with multiple principal frequencies in a small bandwidth and tremor parameters such as amplitude, frequency and bandwidth varying from subject to subject [22,23,24]. Most early techniques used low-pass filters to attenuate physiological tremors, but they had the disadvantage of inherent phase delays that were not conducive to real-time applications [25]. To overcome the time delay problem, some adaptive filtering algorithms such as Fourier Linear Combiner (FLC), weighted-frequency FLC (WFLC) and bandlimited multiple FLC (BMFLC) have been developed, which can adapt to the variation of vibration signal frequency and amplitude [26]. FLC can effectively estimate and eliminate periodic interference at known frequencies, but it cannot handle high frequency signals [27]. WFLC is a modification of FLC for the case where only a single frequency is present in the tremor signal and relies on the estimation of the components of a single frequency; the performance of WFLC degrades if the signal contains multiple dominant frequencies [28]. BMFLC is very accurate for tremor estimation with prior knowledge of the subject’s tremor parameters, yet it is difficult to determine the network structure and search for the optimal solution [29].Thus, tremor filtering models based on the Hilbert transform and Autoregressive (AR), which do not require a priori knowledge, were developed separately, but the Hilbert transform-based model is computationally expensive online, while the AR model treats the tremor signal as a linear Gaussian stochastic process, thus weakening the characteristics of physiological tremor [30,31]. On the other hand, neural network-based adaptive control methods have been used to attenuate physiological tremor. Liu et al. proposed a neural network-based sliding mode control method for robots using a dynamic model obtained by a neural network strategy to approximate the switching gain, which adapts to unknown dynamics and perturbations, but the initial trajectory tracking error is significant because the initial weights of the neural network are chosen randomly [32]. Yang et al. used a broad learning extreme learning machine for the prediction and elimination of tremor in teleoperation, but it is prone to accuracy degradation due to over-fitting [33]. However, the above filtering methods rarely include operator intent into the interaction with the robot for optimal operation.

In this paper, we address the inability of the conventional EKF to cope with the dynamic physiological tremor phenomenon and propose to integrate the sEMG signal into the gain calculation of the EKF so that the improved EKF has the ability of variable gain adaption, which can filter the physiological tremor as well as retain the motion details in the smooth state and realize the personalized teleoperation in the unknown dynamic environment. This paper describes the further development of human upper limb motion capture using wearable sensors in the field of teleoperation. Our contributions are: (1) a method to achieve arm motion capture using IMU in teleoperation is proposed. The advantage of this method is that, considering the misalignment between the rotation axis of the twisted joint and the limb segment, a regression model for the transition from the centroid of the arm to the roll angle at the end of the arm is established to reduce the error of roll angle and can be extended to different individuals. (2) An adaptive extended Kalman filter (EKF) method is proposed to integrate sEMG signals into the control system so that the telerobot can understand the intention of the operator and realize the attenuation of physiological tremor.

## 2. Description of Teleoperation Robot System

A telerobot system consists of two parts: (1) the interactive side and (2) the control terminal. The interactive side of teleoperation is composed of two gForcePro+ armbands, the arm motion capture module and tremor attenuation module, while the control terminal includes an xMate3 Pro robotic arm. After wearing the gForcePro+ armbands on the arm, the arm is perpendicular to the front plane as the initial position. First, the arm joint angles are captured using two IMUs, and the position P_1_ and orientation O_1_ of the arm’s end are obtained by a forward kinematic model. Second, the position P_2_ and orientation O_2_ are obtained by filtering the physiological tremor using the EKF with fused muscle activation. Then, P_2_ and O_2_ are mapped to the robot arm through Cartesian space. Finally, the inverse kinematics of the manipulator are solved, and the joint angles are obtained through trajectory planning to complete the end tracking motion. The overall scheme of the teleoperation system considered in this paper is shown in Figure 1.

In Figure 2, the gForcePro+ armband (Manufacturer: OYMotion Technology Co., Ltd., Shanghai, China) is employed as the interactive side device in the teleoperated robot system. In the system, gForcePro+ armbands are worn on the operator’s upper arm and forearm to measure arm movements. The device includes an 8-channel high-sensitivity sEMG sensor and a 9-axis IMU sensor. Its maximum sEMG sampling frequency is 1000 Hz and IMU sampling is 50 Hz, enabling raw data output. sEMG and IMU data are transmitted via Bluetooth to the interactive side computer.

In Figure 3, the xMate3 Pro robot (Manufacturer: ROKAE (Beijing) Technology Co., Ltd., Beijing, China) is designed with seven degrees of freedom redundant motion, and the robot can achieve the same end position in different configurations. It has a maximum load of 3 kg, a repeatable positioning accuracy of ±0.03 mm, a control frequency of 1000 Hz, support for position control, impedance control and direct torque control and a kinematic and kinetic calculation interface. Motion commands are transmitted via Ethernet to the robot controller.

### 2.1. Human Arm Motion Capture

Since the gForcePro+ armband has both a built-in sEMG sensor and an IMU sensor, the IMU motion capture algorithm can be used to capture the arm joint angle and obtain the raw position and orientation information at the end of the arm through the arm kinematic model. In addition, the IMU motion capture algorithm used in this paper solves the problem of incomplete alignment of the rotation axes of the torsion joints (i.e., shoulder internal/external and elbow internal/external rotation) with the limb segments during rotational movements [34].

#### 2.1.1. Kinematic Model of Human Arm

The human arm consists of three joints comprising seven degrees of freedom: the shoulder joint comprising three DOFs, the elbow joint comprising two DOFs and the wrist joint comprising two DOFs [35]. Using the standard D-H parameter method [36], we can build a kinematic model representing the human arm, as shown in Figure 4. Assuming that the human arm is a seven degrees of freedom tandem robot arm, *l_s_* represents the length of the operator’s shoulder, *l_u_* represents the length of the operator’s upper arm and *l_f_* represents the length of the operator’s forearm.

In the standard DH representation, $T_{i}^{i-1}$ represents the homogeneous coordinate transformation matrix of coordinate systems *i* − 1 to *i*:(1)$$T_{i}^{i-1}=\left[\begin{array}{cccc} \cos\theta_{i} & -\sin\theta_{i}\cos\alpha_{i} & \sin\theta_{i}\sin\alpha_{i} & l_{i}\cos\theta_{i}\\ \sin\theta_{i} & \cos\theta_{i}\cos\alpha_{i} & -\cos\theta_{i}\sin\alpha_{i} & l_{i}\sin\theta_{i}\\ 0 & \sin\alpha_{i} & \cos\alpha_{i} & d_{i}\\ 0 & 0 & 0 & 1 \end{array}\right]$$

Since there are only two IMU sensors on the arm, five joint angles of the arm can be captured, and, assuming that the sixth joint angle is 0, the homogeneous coordinate transformation matrix from the base coordinate system to the end of the arm coordinate system is defined as:(2)$$T_{6}^{0}=T_{1}^{0}T_{2}^{1}T_{3}^{2}T_{4}^{3}T_{5}^{4}T_{6}^{5}$$

#### 2.1.2. IMU Motion Capture Algorithm

Obtain the quaternion data *q*(*w*,*x*,*y*,*z*) from the gForcePro+ armband and transform it into a rotation matrix *R* with the following equation.
(3)$$R=\left[\begin{array}{ccc} 1-2y^{2}-2z^{2} & 2xy-2zw & 2xz+2yw\\ 2xy+2zw & 1-2x^{2}-2z^{2} & 2yz-2xw\\ 2xz-2yw & 2yz+2xw & 1-2x^{2}-2y^{2} \end{array}\right]$$

The rotation matrix of the first IMU (worn on the upper arm) is noted as:(4)$$R_{u}=\left[\begin{array}{lll} r_{11} & r_{12} & r_{13}\\ r_{21} & r_{22} & r_{23}\\ r_{31} & r_{32} & r_{33} \end{array}\right]$$

The rotation matrix of the second IMU (worn on the forearm) is noted as:(5)$$R_{f}=\left[\begin{array}{lll} a_{11} & a_{12} & a_{13}\\ a_{21} & a_{22} & a_{23}\\ a_{31} & a_{32} & a_{33} \end{array}\right]$$
where *u* represents the upper arm coordinate system and *f* represents the forearm coordinate system.

As shown in Figure 5, the angles of the shoulder and elbow joints of the arm are calculated using the following equations.
(6)$$\begin{array}{l} ∠OAB=\arcsin\left(-r_{23}\right)\\ ∠XOB^{\prime }=\mathrm{atan}2\left(r_{13},r_{33}\right)\\ ∠DAE=\mathrm{atan}2\left(r_{21},r_{22}\right)\\ ∠ABC=\arccos\left(a_{21}\cdot r_{21}+a_{22}\cdot r_{22}+a_{23}\cdot r_{23}\right)\\ ∠FBG=\arccos\left(a_{31}\cdot r_{31}+a_{32}\cdot r_{32}+a_{33}\cdot r_{33}\right) \end{array}$$
where ∠*OAB* is the shoulder pitch angle, ∠*XOB’* is the shoulder yaw angle, ∠*DAE* is the shoulder roll angle, ∠*ABC* is the elbow updip angle and ∠*FBG* is the elbow roll angle.

The motion capture theory described above is based on the assumption that the axis of rotation of the torsion joint is perfectly aligned with the limb segment. However, the gforcePro+ armband is worn at the centroid of the arm, which is quite different from the roll angle at the end of the arm, so angle conversion is required. The two gForcePro+ armbands were fixed at the centroid of the arm and the end of the arm, and the roll angle data were collected simultaneously. A polynomial fit was performed to obtain a functional relationship between the two.

Since the human body size varies greatly, it is not a certain definite value but is distributed over a certain range. For individuals of different heights and weights, there are different conversion relationship between the centroid of the arm and the end of the arm for roll angle. To avoid experimenting with each operator, collect the roll angle data and perform a polynomial fit. This paper proposes an empirical formula to facilitate the conversion of the roll angle of the arm’s centroid to the end of the arm for different individuals.

Select an individual as the benchmark individual and establish the roll angle’s conversion relationship between the centroid of the arm and the end of the arm. In order to improve the reusability of the regression model among different individuals, the model should include the effects of height, weight and body mass index (BMI) and assume that their effects on the model are linear. BMI describes the relationship between height and weight, which is significantly correlated with total body fat content, which is calculated as weight (kg)/height squared (m^2^) [37]. According to the inertial parameter standard of the adult human body, the calculation of the body segment centroid involves height and weight. Therefore, the centroid positions of the arm are calculated to represent the influence of height and weight. The calculation formula of the arm’s centroid position is as follows [38]:(7)$$Y=B_{0}+B_{1}X_{1}+B_{2}X_{2}$$
where the values of *B*_0_, *B*_1_ and *B*_2_ refer to Table 1; *X*_1_ is weight (kg) and *X*_2_ is height (mm).

For different individuals, calculate the BMI and the position of the arm’s centroid (*c*) according to the individual’s height and weight, measure the roll angle (*x*) of the arm’s centroid and substitute the following formula to obtain the roll angle of the arm end.
(8)$$u_{end}=\frac{c_{u}}{c_{ub}}u_{b}\left(x\right)$$
where $u_{b}\left(x\right)$ is the polynomial formula of the upper arm of a benchmark individual, $c_{ub}$ is the centroid of the upper arm of a benchmark individual, $c_{u}$ is the centroid of the upper arm of other individuals and $u_{end}$ is the roll angle of the upper arm of other individuals.
(9)$$f_{end}=\frac{BMI}{BMI_{b}}\frac{c_{f}}{c_{fb}}f_{b}(x)$$
where $f_{b}\left(x\right)$ is the polynomial formula of the forearm of a benchmark individual, $c_{fb}$ is the centroid of the forearm of a benchmark individual, $c_{f}$ is the centroid of the forearm of other individuals, $BMI_{b}$ is the body mass index of a benchmark individual, BMI is the body mass index of other individuals and $f_{end}$ is the roll angle of the forearm of other individuals.

### 2.2. Elimination of Physiological Tremor

Physiological tremor is one of the most important human factors affecting the stability and accuracy of teleoperation systems and is unavoidable. As the system is non-linear and the algorithm of the EKF is iterative, it has the advantage of being simple, fast and robust, offering better prospects for real-time applications than filtering methods of neural networks and machine learning. The EKF for state estimation is thus used to filter the position and orientation data. Delays of 30 ms and above may degrade the performance of human-machine control applications [39]. There is an electromechanical delay of approximately 57 ms between muscle activation (sensed by the sEMG) and the onset of movement [40,41]. The use of muscle activation counteracts the inherent electromechanical delays and can provide powerful and intuitive user-driven real-time control of the robot [42]. Aiming to reduce the effect of time delay, the surface sEMG signal of the upper arm is used to estimate muscle activation, which is involved in the calculation of the gain coefficient of the EKF. By introducing muscle activation, the Kalman gain varies with the operator’s upper arm muscle activation. The control gain stabilizes around the optimal control gain when the upper arm’s sEMG signal remains stable, and when the upper arm’s sEMG signal changes significantly, the Kalman gain decreases before the actual tremor occurs, which can solve the inherent time delay problem of the low-pass filter and enhance the filtering effect to filter out the tremor accurately. In addition, the proposed filtering algorithm only processes time-domain signals, which can solve the problem that the frequency adaptation of the adaptive filter cannot handle high-frequency signals [27]. The filter structure of the algorithm is shown in Figure 6. 

#### 2.2.1. Calculation of Muscle Activation

The electromyography is a common method for determining the relative effort and neuromuscular drive of skeletal muscle [43]. Muscle activation can be defined as [44].

In Figure 6, *u*(*k*) is the sum of all channel sEMG signals at moment *k*. In this paper, *u* is measured using an 8-channel gForcePro+ armband.
(10)$$u=\sum_{i=1}^{N}u\left(i\right),i=1,2,3,...,N$$
where *u*(*i*) is the raw sEMG signal of the ith channel.

The raw sEMG signal *u*(*k*) exhibits different characteristics in terms of both frequency and amplitude. This paper uses the mean RMS to determine the amplitude for the raw sEMG signal.
(11)$$\bar{u}(k)=\sqrt{\frac{1}{N}\sum_{j=1}^{N}u_{j}^{2}\left(k\right)},j=1,2,3,...,N$$
where *k* and *N* are the current sampling moment and channel of gForcePro+, respectively.

The exponential moving average filter adopts a form of the recursive algorithm compared to the traditional sliding window averaging filter [45], which only needs to retain the results of the previous moment’s calculation each time with a small resource consumption for the system, smoothing and filtering the RMS feature of the sEMG signal, as follows.
(12)$$\hat{\bar{u}}(k)=\hat{\bar{u}}(k-1)+\frac{1}{G}(z(k)-\hat{\bar{u}}(k-1))$$
where $\hat{\bar{u}}\left(k\right)$ is the posterior estimate of the RMS feature value at moment k, $\hat{\bar{u}}\left(k-1\right)$ is the posterior estimate of the RMS feature value at moment *k* − 1, *z*(*k*) is the measure of the RMS feature value at moment *k* and 1/*G* is the delay factor due to the size of the sliding window, also known as the Kalman gain. The larger *G* is taken, the more pronounced the filtering effect, and the smaller it is, the closer it is to the original signal.

Muscle activation *a*(*k*) contains the sEMG signal and is more stable than the mean RMS feature value [46].
(13)$$a(k)=\frac{e^{A\hat{\bar{u}}(k)}-1}{e^{A}-1}$$
where *A* ∈ (−3,0) is the non-linear shape factor, *A* taken as −3 indicates a highly nonlinear relationship between muscle contraction force and the sEMG signal and *A* taken as 0 indicates a linear relationship. 

Figure 7 shows that the RMS feature is extracted from the blue sEMG signal and then filtered by an exponential moving average filter to obtain the muscle activation.

#### 2.2.2. Variable Gain Extended Kalman Filter Algorithm

In estimation theory, the EKF is a non-linear version of the Kalman filter [47], which linearizes the current mean and estimated covariance. In the EKF, the state transfer model and the observation model are not required to be linear functions of the states but can be differentiable functions [48].

In Figure 6, $x_{k}$ is the position and orientation data for the end of the arm, and ${\overset{ˇ}{x}}_{k}$ is the estimated value of $x_{k}$.
(14)$$x_{k}=\left[\begin{array}{ll} P_{1}(k) & O_{1}(k) \end{array}\right]$$
where $P_{1}\left(k\right)$ and $O_{1}\left(k\right)$ are the position and orientation data, respectively, at the end of the arm at the *k*th moment.

Consider nonlinear random discrete systems:(15)$$\begin{array}{l} x_{k}=f\left(x_{k-1},u_{k}\right)+w_{k}\\ z_{k}=h\left(x_{k}\right)+v_{k} \end{array}$$
where $w_{k}$ and $v_{k}$ are procedure noise and observation noise, both set to zero-mean multivariate Gaussian noise, with covariances $Q_{k}$ and $R_{k}$, respectively, and $u_{k}$ is the control variable. As there are no control inputs, the system input matrix $u_{k}$ = 0.
(16)$$\begin{array}{l} \omega_{k}\sim N\left(0,Q_{k}\right)\\ v_{k}\sim N\left(0,R_{k}\right) \end{array}$$

Due to the non-linear nature of *f*(·) and *h*(·), the motion and observation model needs to be linearized by expanding at the mean of the current state estimate.
(17)$$\begin{array}{c} f\left(x_{k-1},v_{k},w_{k}\right)\approx {\overset{ˇ}{x}}_{k}+F_{k-1}\left(x_{k-1}-{\hat{x}}_{k-1}\right)+w_{k}^{\prime }\\ g\left(x_{k},n_{k}\right)\approx {\overset{ˇ}{y}}_{k}+G_{k}\left(x_{k}-{\overset{ˇ}{x}}_{k}\right)+n_{k}^{\prime } \end{array}$$
where
(18)$$\begin{array}{c} {\overset{ˇ}{x}}_{k}=f\left(x_{k-1},v_{k},0\right),F_{k-1}={\left.\frac{\delta f\left(x_{k-1},v_{k},w_{k}\right)}{\delta x_{k-1}}\right|}_{{\hat{x}}_{k-1},v_{k},0}\\ \omega_{k}^{\prime }={\left.\frac{\delta f\left(x_{k-1},v_{k},w_{k}\right)}{\delta x_{k}}\right|}_{{\hat{x}}_{k-1},v_{k},0}\omega_{k} \end{array}$$
and
(19)$$\begin{array}{c} {\overset{ˇ}{y}}_{k}=g\left({\overset{ˇ}{x}}_{k},0\right),G_{k}={\left.\frac{\delta g\left(x_{k},n_{k}\right)}{\delta x_{k}}\right|}_{{\overset{ˇ}{x}}_{k},0}\\ n_{k}^{\prime }={\left.\frac{\delta g\left(x_{k},n_{k}\right)}{\delta x_{k}}\right|}_{x_{k},0}n_{k} \end{array}$$

Given the past state and the latest input, the statistical properties of the current state and the current observation are:(20)$$\begin{array}{c} p\left(x_{k}|x_{k-1},v_{k}\right)\approx N\left({\overset{ˇ}{x}}_{k}+F_{k-1}\left(x_{k-1}-{\hat{x}}_{k-1}\right),Q_{k}^{\prime }\right)\\ p\left(y_{k}|x_{k}\right)\approx N\left({\overset{ˇ}{y}}_{k}+G_{k}\left(x_{k}-{\overset{ˇ}{x}}_{k}\right),R_{k}^{\prime }\right) \end{array}$$

The prediction equations are as follows.
(21)$$\begin{array}{c} {\overset{ˇ}{P}}_{k}=F_{k-1}{\hat{P}}_{k-1}F_{k-1}^{T}+Q_{k}^{\prime }\\ {\overset{ˇ}{x}}_{k}=f\left({\hat{x}}_{k-1},v_{k},0\right) \end{array}$$

The calculation of the variable Kalman gain coefficient is as follows: as muscle activation varies, Kalman gain also varies. It is crucial to normalize the control gain within a reasonable range influenced by the control strategy and arm stiffness [49]. To achieve filtering performance that varies with muscle activation, the control gain at the kth sampling moment can be expressed as Equation (22).
(22)$$K_{k}=\eta \frac{\left(a_{max}-a_{k}\right)}{\left(a_{max}-a_{min}\right)}\left(K_{max}-K_{min}\right)+K_{min}$$
where *η* ∈ (0,1) is the amplitude factor and $K_{max}$ and $K_{min}$ are the maximum and minimum gains that ensure a steady motion of the robot, adjusted according to experiments. $a_{k}$ is the muscle activation obtained. $a_{max}$ and $a_{min}$ are the maximum and minimum values of muscle activation, respectively. $K_{max}$, $K_{min}$, $a_{max}$ and $a_{min}$ can be obtained experimentally in advance. 

$K_{max}$ and $K_{min}$ have been determined experimentally and are calculated as follows.
(23)$${K={\overset{ˇ}{P}}_{k}G_{k}^{T}\left(G_{k}{\overset{ˇ}{P}}_{k}G_{k}^{T}+R_{k}^{\prime }\right)}^{-1}$$

Since the RMS feature is calculated using the sliding window method, the calculated Kalman gain needs to be refactored. The pre-refactoring Kalman gain is denoted by *K* and the post-refactoring Kalman gain is denoted by *KR*. The offline process of the refactoring Algorithm 1 is shown below.
**Algorithm 1:** Refactor Kalman gain.Input: *K*Output: *KR*00: initialize: Set *KR* = 0, *u* = 001: Set *same_num* = floor(length(*Position_data*)/length(*K*)) 02: **loop**03: **for** *i*=1, 2, …, length(*K*) **do**
04:  **for** *j*=1, 2, …, *same_num* **do**
05:   *KR(u+j)* = EMA(*K(i)*) 06:  **end for**
07:  *u* = *u*+*same_num*
08: **end for**
09: **end loop**

The updated equations are:(24)$$\begin{array}{c} {\hat{P}}_{k}=\left(1-KR_{k}G_{k}\right){\overset{ˇ}{P}}_{k}\\ {\hat{x}}_{k}={\overset{ˇ}{x}}_{k}+KR_{k}\left(y_{k}-g\left({\overset{ˇ}{x}}_{k},0\right)\right) \end{array}$$

## 3. Experiment

The gForcePro+ armband enables motion tracking, but it is difficult to use in practical teleoperation and has two main problems. First, the roll angle measured at the centroid cannot be used for practical robotic arm control. Therefore, in Experiment 1, we established polynomial equations for the reference individuals, and, in order to accommodate different individuals, the corresponding physiological parameters were selected as the influence factors, and a regression model of the roll angle from the centroid of the arm to the end of the arm was established and then compared with the optical tracking system VICON to evaluate the overall accuracy of the IMU system. Second, the position and orientation data are affected by sensor noise and physiological tremors with instability, which are difficult to apply directly to robotic arm control and require filtering of the data. Therefore, in Experiment 2, we compare the EKF method of fusing sEMG with the conventional EKF in order to verify its filtering effect and then apply it to the practical robotic arm control.

### 3.1. Environment of Experiments

In Experiment 1, the subject first performs a correction experiment for the roll angle. As shown in Figure 8a, since the gForcePro+ armband is not very convenient to wear at the centroid of the upper arm, choose 2 cm below the centroid to wear. The gForcePro+ armband can be worn at the centroid when worn on the forearm. The IMU data are collected simultaneously during rotational movements to calculate the roll angle for polynomial fitting. After completing the correction experiment for the roll angle, as shown in Figure 8b, the gForcePro+ armbands are worn on the upper arm and forearm with five Marker points fixed on the arm. The IMU motion capture system and VICON optical capture system are activated simultaneously to capture the motion of the human arm in space to test the performance of the IMU capture algorithm for arm motion capture.

As shown in Figure 9, the Experiment 2 scene is divided into two parts: the interactive side and the control terminal. On the interactive side, the operator wears two gForcePro+ armbands on the upper arm and one on the forearm to capture the position and orientation at the end of the arm for controlling the robot arm and a data glove on the hand to capture the angle of the finger joints for controlling the bionic hand. On the control terminal side, the robot consists of a seven degrees of freedom xMate3 Pro robot arm and a flexible bionic hand, and the robot is used to perform grasping tasks. During grasping, the gForcePro+ armband simultaneously acquires position and orientation data at the end of the arm and sEMG signals from the upper arm to verify the effectiveness of the proposed EKF with variable gain in adaptively attenuating tremor. The object used in the experiment is a 7 cm diameter sphere weighing 100 g.

### 3.2. Experiment 1

The IMU sampling frequency was 50 Hz, and the VICON sampling frequency was 1000 Hz. An adult male with a height of 173 cm and a weight of 63.9 kg was selected as a benchmark. One of the gForcePro+ armbands is placed on the middle of the upper arm/forearm and another is placed on the end of the upper arm/forearm. The roll angle data from both gForcePro+ armbands are collected simultaneously, and a polynomial fit is performed to obtain a functional relationship between the two. Figure 10 shows the relationship between the order of fit and RMSE. For the forearm roll angle correction, the RMSE reaches a small value when the polynomial order is 2, so increasing the polynomial order is not significant at this moment; for the upper arm roll angle correction, the RMSE reaches a small value when the polynomial order is 3, so increasing the polynomial order is not significant at this moment.

The polynomial fit equation between the mid and end rotation angles of the upper arm is obtained as Equation (25). The fitted curve is shown in Figure 11.
(25)$$u_{bend}(x)=7.14\times 10^{-4}x^{3}-0.068x^{2}+3.566x+1.999$$
where *x* is the roll angle in the middle of the upper arm and $u_{bend}\left(x\right)$ is the roll angle at the end of the upper arm.

The polynomial fit equation between the mid and end rotation angles of the forearm is obtained as Equation (26). The fitted curve is shown in Figure 12.
(26)$$f_{bend}\left(x\right)=0.0155x^{2}+2.665x-0.479$$
where *x* is the roll angle in the middle of the forearm and $\;f_{bend}\left(x\right)$ is the roll angle at the end of the forearm.

In order to evaluate the effectiveness of the regression model, validation experiments were carried out. Ten healthy male volunteers (mean, 25.6 ± 0.8 years old) participated in the experiment. The regression model requires two human physiological parameters: the centroid (c) of the arm and BMI. These parameters are calculated according to the empirical formula of China Standardization Administration [38]. Table 2 lists the experimental participants’ physiological data involved in the experiment. Each subject substituted the roll angle in the middle of the arm into the regression model to obtain the predicted roll angle at the end, which was compared with the actual measured rolling angle at the end for error analysis. The average RMSE of the roll angle regression model of the forearm is 3.52 ± 3.16, and that of the upper arm is 3.61 ± 2.29.

After correction of the roll angle, the modified arm motion capture method needs to be compared with the VICON optical capture system to assess the capture accuracy. The right arm is placed in front of the body and kept horizontal, two gForcePro+ armbands are connected separately, marker points are fixed on the arm, the IMU is calibrated, the VICON optical capture system is activated to start the motion and the motion data is collected simultaneously. RMSE and R^2^ are used to quantify performance.

The arm joint angles captured by the gForcePro+ armband are compared to those captured by the VICON optical capture system. Figure 13 shows the angles measured by the system and the angles measured by VICON, with close trajectories between the two. In particular, the shoulder and elbow joints were captured with the expected accuracy for the roll angle, indicating that the roll angle correction is effective. The error results are shown in Table 3, with a system RMSE of 4.837 ± 1.433 and an R^2^ of 0.834 ± 0.073.

### 3.3. Experiment 2

The IMU sampling frequency was 50 Hz, the sEMG signal sampling frequency was 500 Hz and the non-linear shape factor *A* was −0.01. The filter delay factor *G* of exponential moving average filter was 5, the $K_{max}$ of EKF was 0.618 and the $K_{min}$ of EKF was 0.068.

To verify the effectiveness of the proposed variable gain filtering method, we collected position and orientation data at the end of the arm during the experiment. Using the Z-direction movement as an example, the same segment of data is tested for tremor attenuation in three modes: (1) fixed high-gain mode; (2) fixed low-gain mode; and (3) variable gain mode.

Figure 14 shows the arm trajectory tracking in the Z direction at high gain mode, indicating that the EKF with fixed high gain can capture the motion details well but cannot effectively filter the tremor. Figure 15 shows the arm trajectory tracking in the Z-direction at low gain mode. It can be shown that the EKF with a fixed low gain can filter the tremor well, but the capture of motion details is weak, and there is a delay effect. This is because physiological tremor is generated randomly and unpredictably, and a fixed gain cannot switch the operating intensity of the filter between non-tremors and tremors.

As shown in Figure 16, in the case of trajectory tracking in the Z-direction, the proposed variable gain-based EKF method performs best compared to the other two modes. The variable gain filtering mode can capture motion details well in the case of smoothness and adaptively adjust the gain in the presence of tremor according to the change in muscle activation, achieving an excellent filtering effect on tremor and preservation of motion details. The results show that the proposed filtering algorithm effectively counters the uncertain dynamics during teleoperation. The reason is that the Kalman gain is constantly changing due to muscle activation and is pre-sensed by the sEMG signal before the onset of tremor, and the filter enables dynamic switching of filter performance in both tremor and non-tremor situations.

Teleoperation experiments with tremor and tremor attenuation are carried out separately to verify the effectiveness of the algorithm for real-time applications.

Teleoperation with tremor: the reference trajectory is generated by the gForcePro+ armband worn by the operator and the actual trajectory is captured by the xMate3 Pro robotic arm. The experimental results are shown in Figure 17. The results show a significant tremor effect throughout the tracking process (0–5 s), and the trajectory tracking performance of the robot is degraded by the tremor, which is not conducive to the smooth operation of the robotic arm.

Teleoperation with tremor attenuation: In this experiment, a variable gain EKF is used to process the trajectory data. Figure 18 shows how the filter’s performance varies with muscle activation and the advantage of the variable gain EKF in response to uncertainty tremor. Subplot a shows that, in the case of a smooth state, the gain coefficient KR is at a high value and the filter only plays a smoothing role. Subplot b shows that, in the case of significant tremor, the gain coefficient KR decreases rapidly, the filter performance is enhanced, the tremor is attenuated and the trajectory becomes smooth without delay. The actual trajectory of the robot follows the change of the reference trajectory well, which reflects the better performance of the proposed method in terms of real-time tremor attenuation.

Table 4 shows the error analysis of the teleoperation in the case of tremor and tremor attenuation, with RMSE and goodness of fit (R^2^) as the analysis metrics. Using the proposed filtering algorithm, the trajectory error between the operator and robot decreased, where RMSE decreased by 0.0009 m, R^2^ improved by 1.50% and the trajectory tracking performance was enhanced. The reason for this is that the physiological tremor is filtered out, the motor of the robotic arm does not have to start and stop frequently and the motor motion is smooth.

## 4. Discussion

In Experiment 1, we corrected the roll angle during the rotational movement of the arm in the process of capturing the position and orientation at the end of the arm using the IMU sensor. Table 2 shows the RMSEs of the ten subjects who participated in the cross-roll angle correction experiment. The mean RMSEs of the upper arm and forearm were 3.52 and 3.61, respectively, indicating that the regression model was accurate, but the performance was poorer in some individuals due to human variability, reaching more than 9°, indicating that the model has room for improvement. A comparison with the VICON optical capture system shows that the proposed IMU system is able to estimate the angle of the arm joint stably for the precise operation of teleoperation tasks.

In Experiment 2, the position and orientation at the end of the arm calculated by the IMU sensor can be unstable due to the inherent noise of the sensor and the occasional tremor of the human arm. This can affect the accuracy of the teleoperated robot’s operation. In this paper, an EKF fusing sEMG is used to attenuate physiological tremor. A comparison was made between the traditional fixed-gain EKF and the variable-gain EKF on the performance of arm trajectory tracking. From Figure 14 and Figure 15, it can be seen that neither fixed high-gain nor fixed low-gain can handle dynamic uncertainty tremors well and they both have their drawbacks. Therefore, a variable gain filtering method is essential for robotic teleoperation. From Figure 16, we can observe that the Kalman gain of the proposed method varies with muscle activation. Compared to a fixed low-gain or high-gain, the method can adaptively filter the steady-state and tremor state, achieving good tremor filtering while retaining motion details. Figure 17 shows the raw, unfiltered data with multiple jumps and tremors that are detrimental to the operation of the robot arm motor and are prone to excessive acceleration and emergency stops. Figure 18 shows the filtering process of the variable-gain EKF. Subplots a and b show the filtering of the tremor with different muscle activation, which indicates different filtering performance. These present the benefits of variable gain filtering. Table 4 shows the real-time tracking performance in the presence of tremor and tremor attenuation. Compared to the condition of having tremors, the RMSE for tremor attenuation is smaller, indicating that the method can reduce the error caused by the tremor. Moreover, R^2^ closer to 1 represents better tracking performance of the robot in the case of tremor attenuation.

## 5. Conclusions

This paper presents a method of IMU motion capture with adaptive filtering in the teleoperation system. The method captures the motion of the human upper limbs and provides the position and direction information of the end of the arm, which can be mapped onto the end of the robot after adaptive filtering to complete the control. The main conclusions are summarized below:The end of arm positioning in motion capture was completed by using two gForcePro+ armbands. A regression model was developed for the roll angle transition between the centroid of the arm and the end of the arm. This model can be generalized to different individuals.Based on the changes in muscle activity during human arm movements, a variable gain extended Kalman filtering method is proposed to filter physiological tremors and device noise. Furthermore, it retains smooth state details to facilitate precision operations.A comparison experiment with an optical tracking system was designed to verify the accuracy of IMU motion capture. The results show that the roll angle error is at the same level as the other angles, with an average error of 4.837° for all angles, indicating that the IMU system can be used for teleoperation.A teleoperation experiment of tremor attenuation was conducted to test the effectiveness of the proposed adaptive filtering method. The results show that the proposed filtering method can reduce the trajectory error between the operator and robot, in which the RMSE is reduced by 0.0009 m.

## Figures and Tables

**Figure 1 sensors-22-03353-f001:**
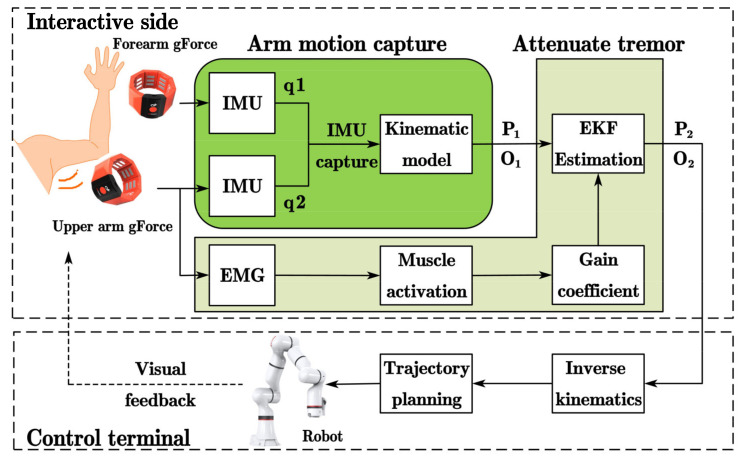
Principle of the teleoperated robot system.

**Figure 2 sensors-22-03353-f002:**
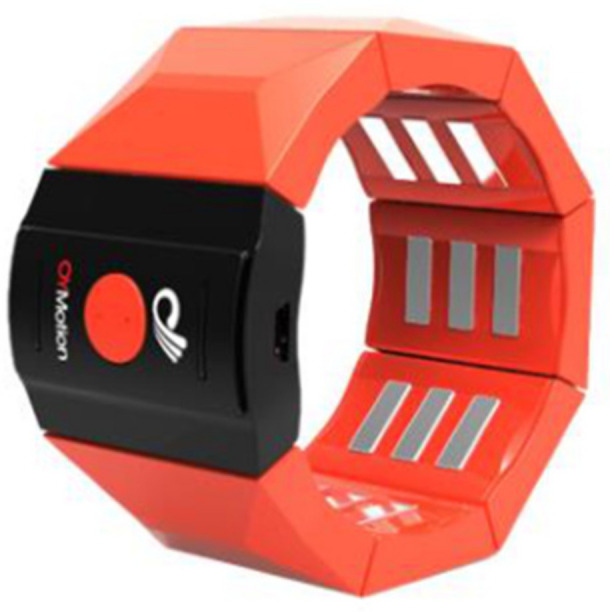
gForcePro+ armband.

**Figure 3 sensors-22-03353-f003:**
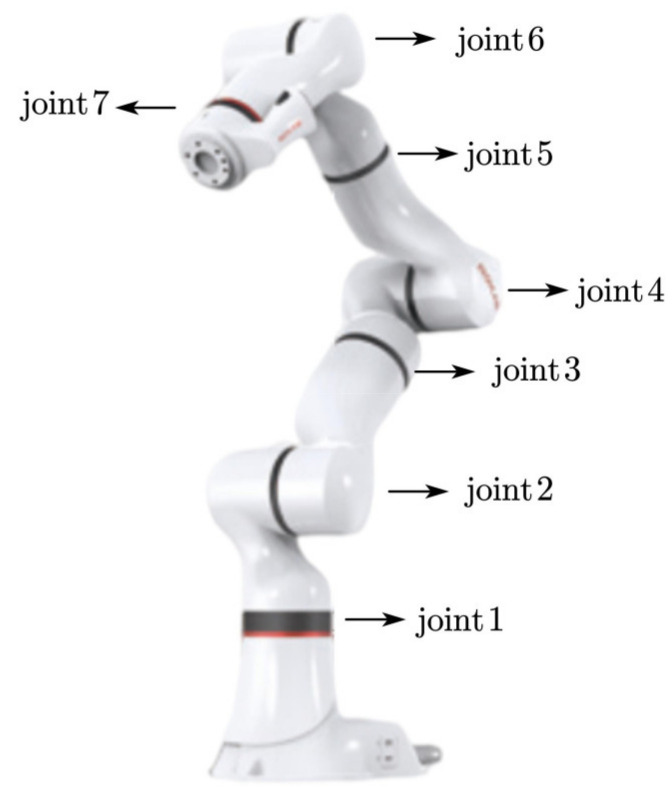
xMate3 Pro robot.

**Figure 4 sensors-22-03353-f004:**
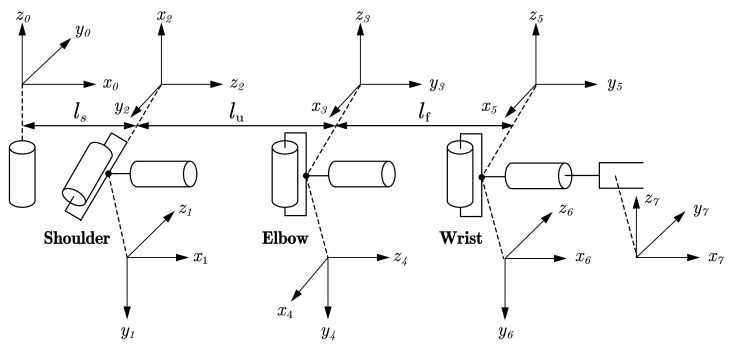
Kinematic model of the human arm.

**Figure 5 sensors-22-03353-f005:**
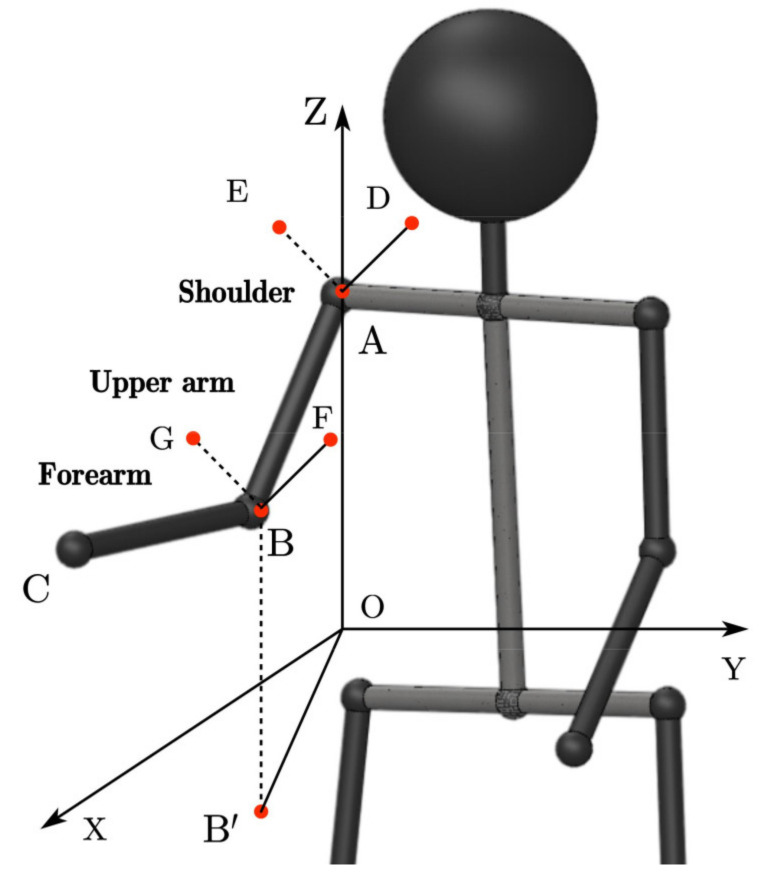
Description of arm joint angle.

**Figure 6 sensors-22-03353-f006:**
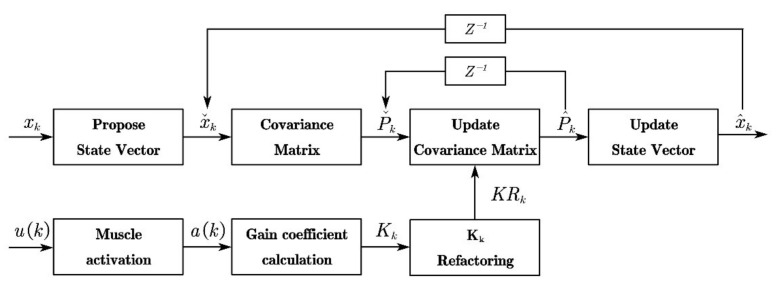
Structure of the variable gain EKF fusing muscle activation.

**Figure 7 sensors-22-03353-f007:**
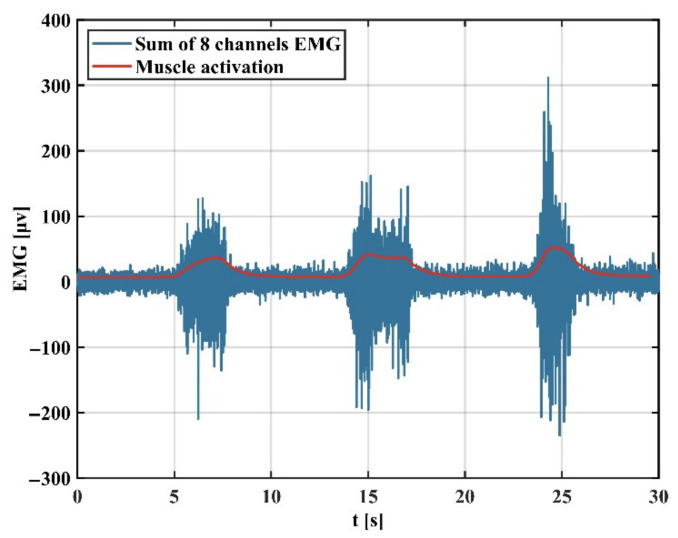
sEMG signal and its corresponding muscle activation.

**Figure 8 sensors-22-03353-f008:**
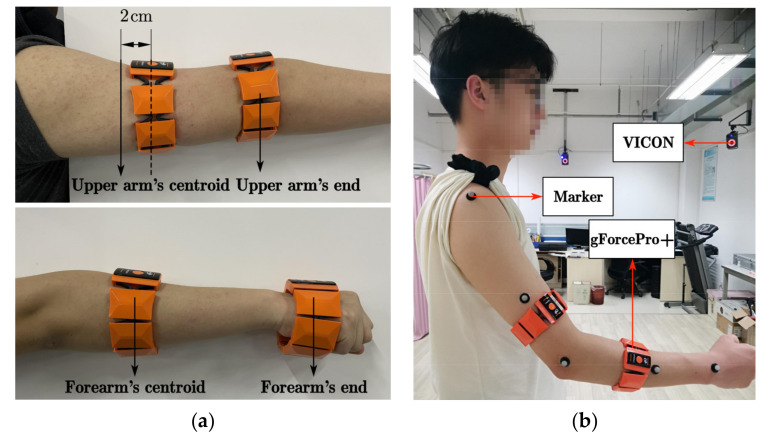
Environments of Experiment 1. (**a**) Correction experiment for the roll angle. (**b**) Angles estimation with the two gForcePro+ systems compared to the angle estimated by the VICON.

**Figure 9 sensors-22-03353-f009:**
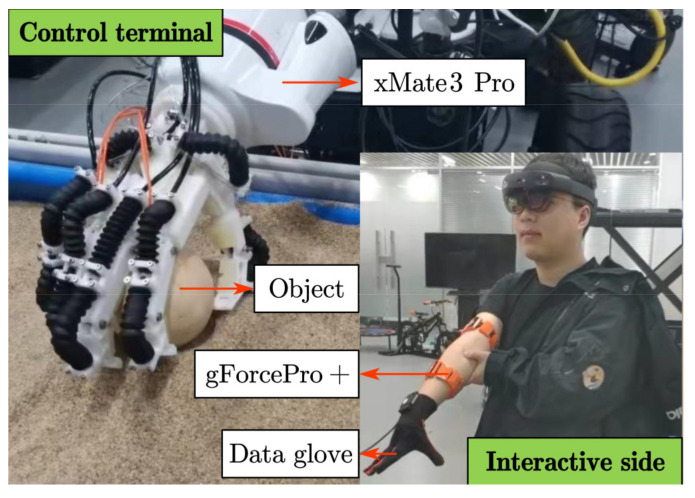
Environments of Experiment 2.

**Figure 10 sensors-22-03353-f010:**
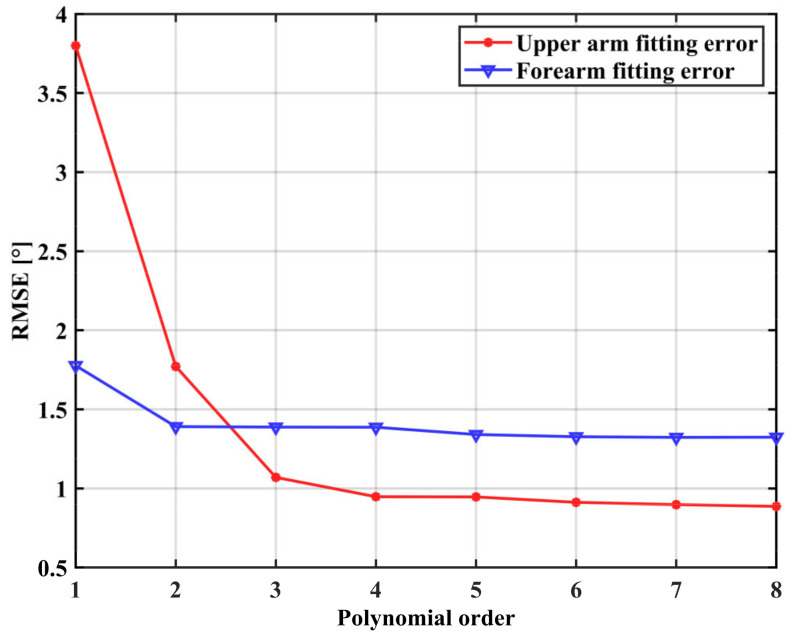
The relationship between the fitting order and RMSE of the benchmark individual.

**Figure 11 sensors-22-03353-f011:**
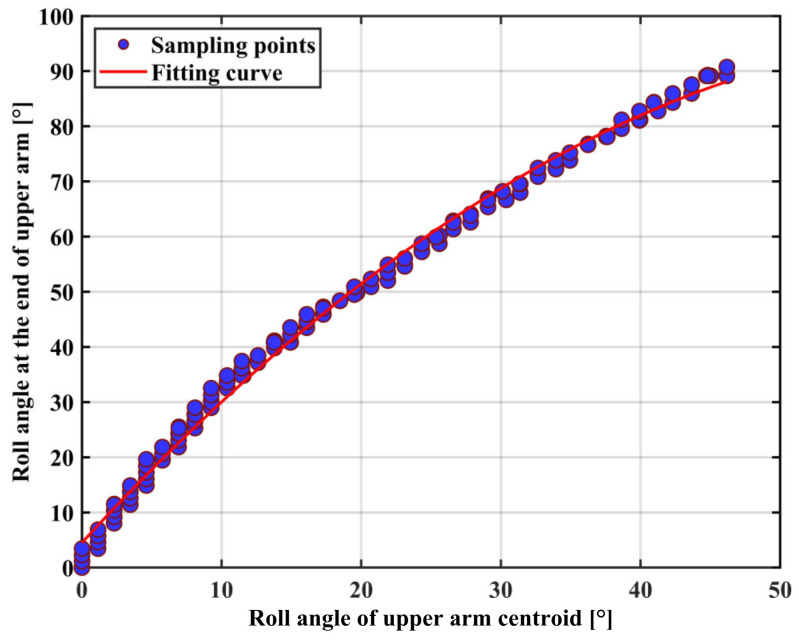
Fitting of the roll angle between the centroid and the end of the upper arm.

**Figure 12 sensors-22-03353-f012:**
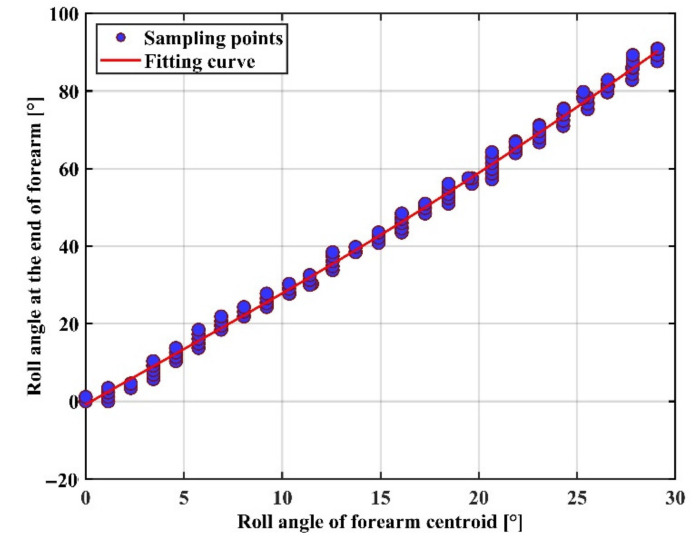
Fitting of the roll angle between the centroid and the end of the forearm.

**Figure 13 sensors-22-03353-f013:**
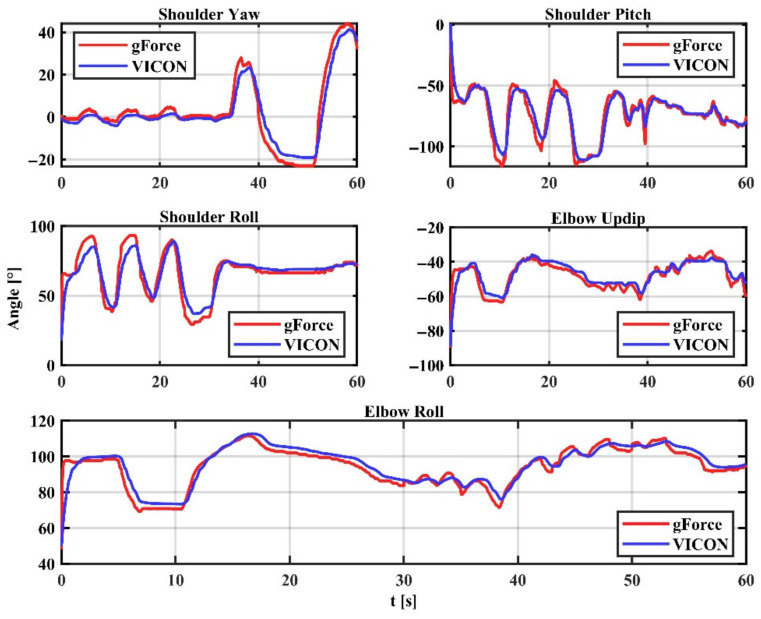
Comparison of gForcePro+ and VICON.

**Figure 14 sensors-22-03353-f014:**
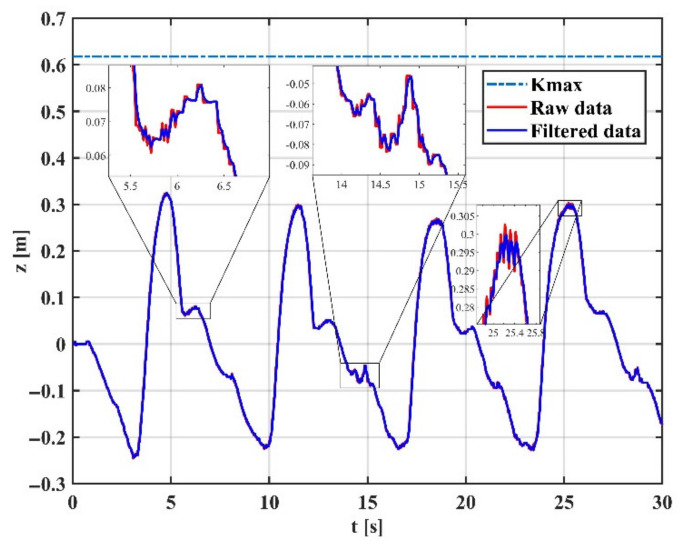
Motion filtering in high gain mode.

**Figure 15 sensors-22-03353-f015:**
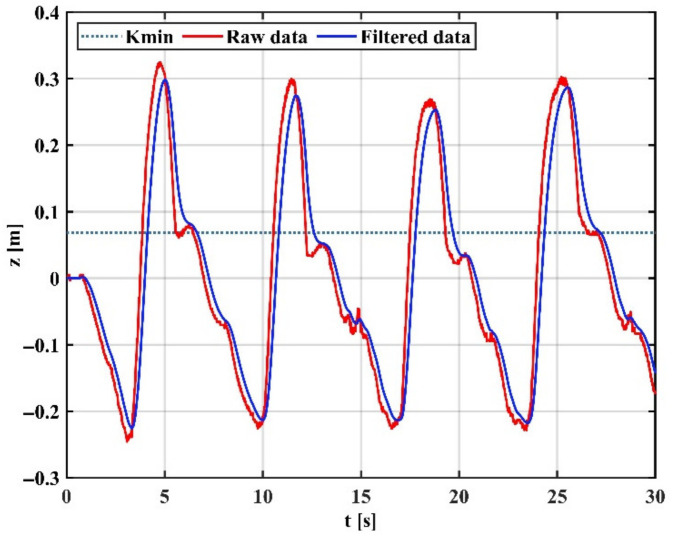
Motion filtering in low gain mode.

**Figure 16 sensors-22-03353-f016:**
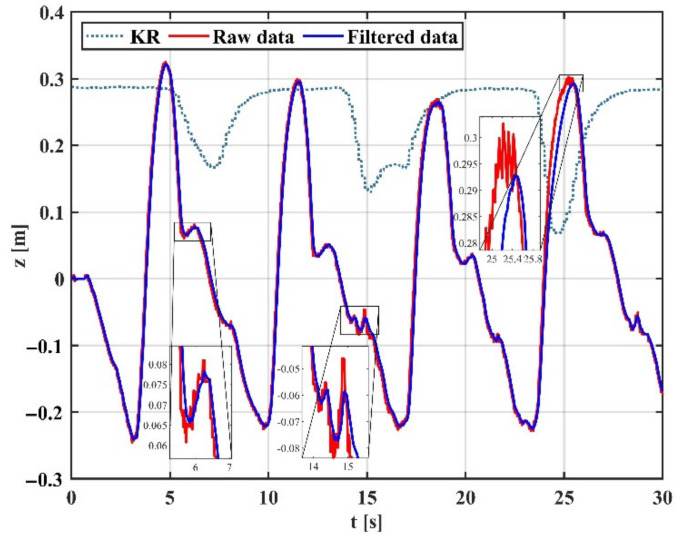
Motion filtering in variable gain mode.

**Figure 17 sensors-22-03353-f017:**
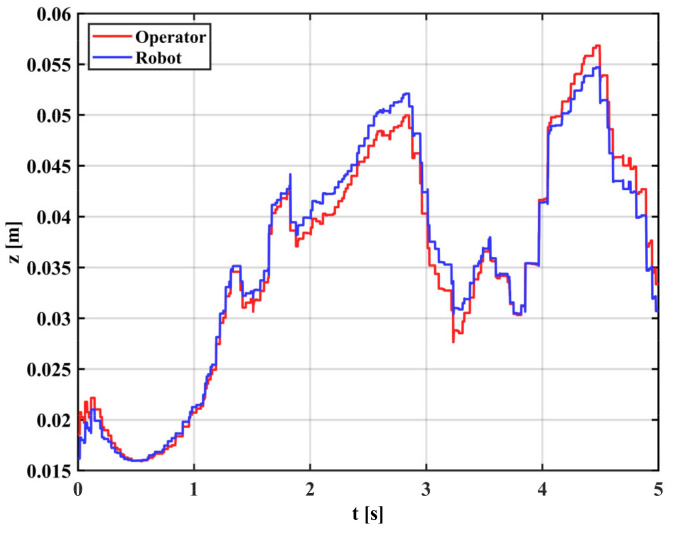
Operator and robot’s trajectory with tremor.

**Figure 18 sensors-22-03353-f018:**
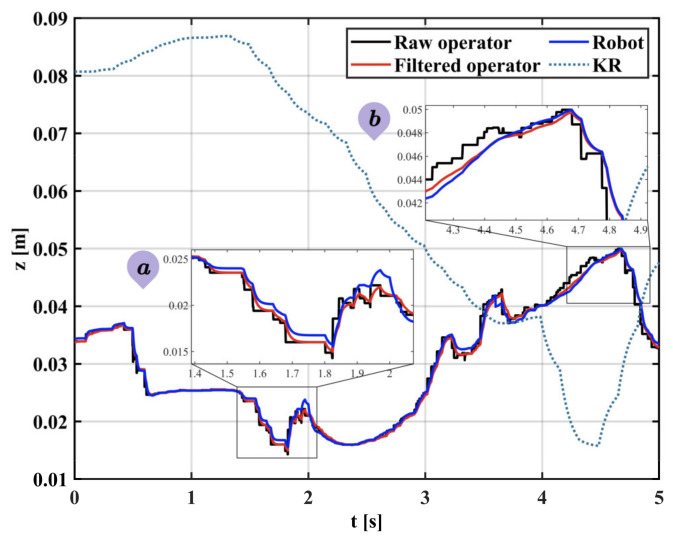
Operator and robot’s trajectory after tremor attenuation. Subplot (**a**) is the case of filtering in the smooth state and subplot (**b**) is the case of filtering in the tremor state.

**Table 1 sensors-22-03353-t001:** Coefficient of weight and height in men.

Body Segment	Constant Term of the Regression Equation	Regression Coefficient of Weight	Regression Coefficient of Height
Centroid of upper arm	15.15	0.16	0.080
Centroid of forearm	12.94	0.45	0.054

**Table 2 sensors-22-03353-t002:** Physiological data of ten experimental subjects.

Subjects	Height /cm	Weight /kg	c of Forearm /cm	c of Upper Arm/cm	BMI/kg/m^2^	RMSE of Roll Angle of Forearm	RMSE of Roll Angle of Upper Arm
Benchmark	173	63.9	135.115	163.774	21.4	1.07	1.39
S1	171	77.0	139.930	164.270	26.3	2.05	0.27
S2	175	74.9	141.145	167.134	24.5	3.00	6.32
S3	177	70.8	140.380	168.078	22.6	0.35	1.74
S4	173	72.0	138.760	165.070	24.1	2.96	5.34
S5	176	81.0	144.430	168.910	26.1	3.55	2.82
S6	174	66.1	136.645	164.926	21.8	1.59	3.40
S7	189	76.3	149.335	178.558	21.4	9.87	2.65
S8	178	67.0	139.210	168.270	21.1	0.17	3.69
S9	167	59.5	129.895	158.270	21.3	3.48	7.86
S10	168	67.5	134.035	160.350	23.9	8.18	2.03
Mean ± SD	174.80 ± 6.18	71.21 ± 6.36	139.38 ± 5.33	166.384 ± 5.499	23.3 ± 2.0	3.52 ± 3.16	3.61 ± 2.29

**Table 3 sensors-22-03353-t003:** Arm estimation accuracy using our approach versus the VICON system.

	Sh.Yaw	Sh.Pitch	Sh.Roll	EL.Updip	EL.Roll	Mean ± SD
**RMSE**	4.079	6.264	6.501	3.437	3.905	4.837 ± 1.433
**R^2^**	0.920	0.869	0.745	0.772	0.865	0.834 ± 0.073

**Table 4 sensors-22-03353-t004:** Error analysis of teleoperation in the case of tremor and tremor attenuation.

Mode	Direction	RMSE	R2
With tremor	z-Direction	0.0015	0.9819
With tremor attenuation	z-Direction	0.0006	0.9969

## Data Availability

The data presented in this study are available on request from the corresponding author.

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
