# Peer review of "IMU Motion Capture Method with Adaptive Tremor Attenuation in Teleoperation Robot System"

_sensors, 2022, doi:10.3390/s22093353_

Round 1

Reviewer 1 Report

This paper presents a motion capturing method for IMU-based motion capture system with an adaptive extended Kalman filter method. The objective is clear, but the originality of this method is not clear. In the introduction, the authors introduced a lot of basic topics of the teleoperation, motion capture system, and IMU sensors. However, directly related researched on the filtering techniques by other researchers have not been introduced. From this reason, the originality of this method is not clear. What is the major originality in terms of the filtering method? A major revision should be done before publication. In addition, the following points should be solved.

Minor points

  1. The maker and model number of the gForce armband should be presented.
  2. In the same way, the maker and model number of the ROKAE robot should be presented.
  3. The master/slave terminology is now unfavorable expression and it's preferable to use alternative words.
  4. The meaning of the figure 5 is not clear. What is the origin? Which plane is the frontal plane of human?
  5. The validity of the equation 7 and table 1 is not clear. How much is the coefficient of determination? And what is the limitation of this approximation?
  6. Equation 11 is not a definition of the normal RMS.
  7. Figure 7 shows strange results. The raw signal shows a negative drift, but the result shows a positive drift. Why?
  8. How did you adjust the gain G and A in Eq.12 and 13?

Reviewer 2 Report

The authors introduce their work about IMU (inertial measurement units) motion capture with adaptive filtering in the teleoperation system. The fundamental theories are supported by the following experiments. The only concern is the evaluation and the analysis of the relation between the co-bot and the arm motion. The muscles’ behavior decides the pose and the location of the co-bot. How do you control the procedures from one pose and location to the target ones? I hope that the authors may add these contents to the revised draft.

Round 2

Reviewer 1 Report

All questions have been clearly answered.